# Effect of *Aruncus dioicus* var. *kamtschaticus* Extract on Neurodegeneration Improvement: Ameliorating Role in Cognitive Disorder Caused by High-Fat Diet Induced Obesity

**DOI:** 10.3390/nu11061319

**Published:** 2019-06-12

**Authors:** Su Bin Park, Jin Yong Kang, Jong Min Kim, Seon Kyeong Park, Seul Ki Yoo, Uk Lee, Dae-Ok Kim, Ho Jin Heo

**Affiliations:** 1Division of Applied Life Science (BK21 plus), Institute of Agriculture and Life Science, Gyeongsang National University, Jinju 52828, Korea; tbsk5670@naver.com (S.B.P.); kangjy2132@naver.com (J.Y.K.); myrock201@naver.com (J.M.K.); tjsrud2015@naver.com (S.K.P.); ysyk9412@naver.com (S.K.Y.); 2Division of Special Forest Products, National Institute of Forest Science, Suwon 16631, Korea; rich26@korea.kr; 3Department of Food Science and Biotechnology, Kyung Hee University, Yongin 17104, Korea; dokim05@khu.ac.kr

**Keywords:** neurodegeneration, metabolic syndrome, insulin resistance, high-fat diet, *Aruncus dioicus* var. *kamtschaticus*

## Abstract

This study was performed to estimate the possibility of using an ethyl acetate fraction from *Aruncus dioicus* var. *kamtschaticus* (EFAD) on metabolic syndrome that is induced by a high-fat diet (HFD). It was demonstrated that EFAD suppresses lipid accumulation and improves insulin resistance (IR) caused by Tumor necrosis factor alpha (TNF-α) in *in-vitro* experiments using the 3T3-L1 cell. In *in-vivo* tests, C57BL/6 mice were fed EFAD at 20 and 40 mg/kg body weight (BW) for four weeks after the mice were fed HFD for 15 weeks to induce obesity. EFAD significantly suppressed the elevation of BW and improved impaired glucose tolerance in obese mice. Additionally, this study showed that EFAD has an ameliorating effect on obesity-induced cognitive disorder with behavioral tests. The effect of EFAD on peripheral-IR improvement was confirmed by serum analysis and western blotting in peripheral tissues. Additionally, EFAD showed an ameliorating effect on HFD-induced oxidative stress, impaired cholinergic system and mitochondrial dysfunction, which are interrelated symptoms of neurodegeneration, such as Alzheimer’s disease and central nervous system (CNS)-IR in brain tissue. Furthermore, we confirmed that EFAD improves CNS-IR by confirming the IR-related factors in brain tissue. Consequently, this study suggests the possibility of using EFAD for the prevention of neurodegeneration by improving metabolic syndrome that is caused by HFD.

## 1. Introduction

Obesity is globally prevalent as diets change to “Western diets” that consist of fat (unhealthy oils), sugar, and animal resource foods. With the increase in obesity, the World Health Organization estimates that diabetes will increase to about 522 million people by 2035 [1]. Obesity is a metabolic syndrome that is caused by an imbalance between the ingested diet and energy consumed, and it has recently been linked to adipose tissue dysfunction [2,3,4]. Adipose tissue is a complex endocrine and immune organ that affects many metabolic functions that are related to vascular-stromal fractions (such as macrophages, fibroblasts, endothelial cells, and pre-adipocytes) and adipokine secretion, and not inert tissue that only functions as energy storage [2]. Adipose tissue dysfunction that is caused by excessive accumulation of fat induces low-grade inflammation, such as a chronic pro-inflammatory reaction. Consequently, insulin resistance (IR) that is related to glucose metabolism dysfunction occurs [3,4,5]. IR causes metabolic disorders by inducing an increase in blood glucose and insulin levels due to decreased susceptibility to insulin in muscle, liver, and adipocytes [5]. It is known that IR is a major symptom of type 2 diabetes (T2DM). Recently, it has been reported that chronic hyperglycemia and hyperinsulinemia due to peripheral-IR cause central nervous system (CNS)-IR, which in turn results in cognitive impairment in the brain, leading to Alzheimer’s disease (AD) [6].

AD is a neurodegenerative disease that is characterized by oxidative stress, impaired mitochondrial function, inflammatory response, cholinergic system changes, extracellular amyloid-β (Aβ) plaque formation, and intracellular neurofibrillary tangles [6,7]. It is reported that the number of dementia patients will exceed 1.9 million by 2040 [8]. Recently, AD has been referred to as type 3 diabetes (T3DM), due to the mechanism of sharing with T2DM for IR and hyperglycemia. Additionally, it is reported to contribute to the increased incidence of AD as a complication of T2DM [6].

Large amounts of phenolics, which are secondary metabolites of plants, such as caffeic acid, benzoic acids, cinnamic acid, and flavonoids, are contained in in food, and they are known to be able to prevent various diseases in their role as excellent antioxidants [9]. Recently, these phenolics have been reported to have anti-obesity effects by controlling energy homeostasis by promoting lipid β-oxidation of peripheral tissues and increasing the free fatty acid circulation [10]. Additionally, it also has a role as potential anti-diabetic functional ingredients by controlling postprandial hyperglycemia through its strong α-glucosidase inhibitory effect [11]. Therefore, there is growing interest in the complex physiological activities of phenolics, such as anti-obesity and anti-diabetic, as well as antioxidant activity. *Aruncus dioicus* var. *kamtschaticus* is wild herb that is distributed in Northeast Asia such as Korea, Japan, and China, recently, a variety of research results have demonstrated the physiological activity of the *Aruncus dioicus* var. *kamtschaticus*. In *in vitro* study, it is reported that the extract of *Aruncus dioicus* var. *kamtschaticus* protects DNA damage and inhibits the NO activity as well as levels of pro-inflammatory cytokines mRNA expressions in lipopolysaccharide (LPS) stimulated RAW 264.7 cells [12]. It is confirmed that *Aruncus dioicus* var. *kamtschaticus* had hypoglycemic and hypolipidemic effects in an animal model of type 2 diabetes in the animal model experiment [11]. In addition, it was confirmed in a previous study that the ethyl acetate fraction from *Aruncus dioicus* var. *kamtschaticus* (EFAD) has an excellent effect on mitochondrial dysfunction based on the superb antioxidant activity of dicaffeoyl glucose isomers [13,14]. Therefore, in this study, we tried to clarify the relationship between peripheral tissues and CNS by *in-vitro* and animal model experiments, and estimate the potential of EFAD in obesity-induced diabetes and further cognitive impairment.

## 2. Materials and Methods 

### 2.1. Chemicals

Acetylthiocholine iodide, bovine serum albumin (BSA), ethylene glycol tetraacetic acid (EGTA), dimethyl sulfoxide (DMSO), thiobarbituric acid (TBA), Triton X-100, Tween 20, L-Glutathione, tetrachloro-1,1,3,3-tetraethylbenzimidazolylcarbocyanine iodide (JC-1), 5,5′-dithiobis(2-nitrobenzoic acid) (DTNB), 2′,7′-dichlorofluorescein diacetate (DCF-DA), metaphosphoric acid, and all other chemicals were obtained from Sigma-Aldrich Chemical Co. (St. Louis. MO, USA). A superoxide dismutase (SOD) assay kit was purchased from Dojindo Molecular Technologies (Kumamoto, Japan) and an adenosine triphosphate (ATP) assay kit was purchased from Promega Corporation (Madison, WI, USA). The primary antibodies [β-actin (sc-69879), phosphorylated-insulin receptor substrate (p-IRS)(ser307), phosphorylated-tau (p-tau)(ser404) (sc-12952), Acetylcholinesterase (AChE) (sc-373901), phosphorylated-protein kinase B (p-Akt) (sc-514032), amyloid-β (Aβ) (sc-28365), and insulin degradation enzyme (IDE) (sc-393887)] were purchased from Santa Cruz Biotechnology (Santa Cruz, CA, USA). Other primary antibodies (Tumor necrosis factor alpha (TNF-α) (#3707), and phosphorylated-AMP activated kinase (p-AMPK) (#2535)) and secondary antibodies (anti-mouse (7076S) and anti-rabbit (7074S)) were purchased from Cell Signaling Technology (Danvers, MA, USA).

### 2.2. Sample Preparation

*Aruncus dioicus* var. *kamtschaticus* was purchased from Taebaek Ssamchae Village (Taebaek, Korea) and verified by the National Institute of Forest Science. The dried *Aruncus dioicus* var. *kamtschaticus* was extracted with 80% ethanol for 2 h 30 min. After being concentrated, fractionated with n-haxane, chloroform, ethyl acetate, and distilled water sequentially, EFAD was used in this experiment [14].

### 2.3. T3-L1 Cell Culture and Differentiation

3T3-L1 adipocytes were cultured and maintained in Dulbecco’s modified Eagle’s medium (DMEM) containing 10% bovine calf serum (CS) and 1% antibiotics, in a 5% CO_2_ atmosphere at 37 °C. The cells were incubated on a plate until they were merged, and were then replaced with MDI medium (DMEM medium containing 10% fetal bovine serum (FBS) and MDI solution (0.5 mM IBMX, 0.5 μM dexamethasone, 10 μg/mL insulin)) for differentiation. After being incubated for three days, insulin medium (DMEM medium containing 10% FBS and 10 μg/mL insulin) was treated for six days (medium replaced three times for 6 days) [15].

#### 2.3.1. Cell Viability (MTT Assay)

Cell viability was evaluated by MTT assay. 3T3-L1 preadipocytes were plated into 96-well microtiter plates at a density of 1 × 104 cells/well. After 24 h, the cells were treated with EFAD for 24 h. MTT solution (5 mg/mL in phosphate buffered saline; PBS) was added to each well and then the cells were incubated for 3 h. The medium was removed and DMSO was added to dissolve the MTT formazan. The absorbance was measured at 570 nm while using a microplate reader 

#### 2.3.2. Lipid Accumulation (Oil Red O Staining)

Each concentration of EFAD were added in the medium during the differentiation process, and the medium was removed after the differentiation process. Then, 10% formalin was added to fix the cells for 1 h. After that, the formalin was removed, it was washed with 60% isopropanol and the 60% isopropanol was completely evaporated in a hood. It was treated with Oil Red O dye for 15 min., rinsed with distilled water, and placed in 100% isopropanol for 10 min. Each absorbance at 500 nm was measured [16].

#### 2.3.3. TNF-α Induced Insulin Resistance in 3T3-L1 Cells

After the completion of the differentiation process, the cells were treated for 6 h at different concentrations (50, 100 μg/mL) of EFAD in the insulin medium. Subsequently, TNF-α was added to the medium at a treatment concentration of 20 ng/mL for a day (24 h) [17]. After cell lysis, a pretreatment process for western blotting analysis was conducted [15].

### 2.4. Animal Model

In this study, C57BL/6 mice (male, four weeks) were purchased from Samtako (Osan, Korea) and used. The mice were housed two to three per cage in a room with laboratory conditions (12 h light-dark cycle, 55% humidity, and a 22 ± 2 °C temperature). The mice had free access to water and food and had a weeklong adaptation period before the comparison diet. After one week (the adaptation period), the mice were randomly divided into four groups (*n* = 7). The control group was fed a normal diet (including carbohydrate (70 kcal%/g), protein (20 kcal%/g), and fat (10 kcal%/g), 3.85 kcal/g), and the rest of the groups were fed an high-fat diet (HFD) (including carbohydrate (20 kcal%/g), protein (20 kcal%/g) and fat (60 kcal%/g), 5.24 kcal/g) for 15 weeks until a hyperglycemic state was induced. After 15 weeks of being fed an HFD, the two groups started to orally consume EFAD at 20, 40 mg/kg of the averages of body weight (BW), respectively, for four weeks (HFD+EFAD20, HFD+EFAD40). The same volume of water was orally administered to the control and HFD group. BW, food intake, and fasting blood glucose from the tail vein were measured weekly. All of the animal experiments were carried out in accordance with the Policy of the Ethical Committee of the Ministry of Health and Welfare, Republic of Korea, and procedures were approved by the Institutional Animal Care and Use Committee (IACUC) of Gyeongsang National University (certificate: GNU-131105-M0067) [18].

### 2.5. Fasting Blood Glucose Measurement and Intraperitoneal Glucose Tolerance Test (IPGTT)

The fasting blood glucose levels were measured by using an Accu-Chek glucose meter (Roche Diagnostics Australia Pty. Ltd., Castle Hill, Australia) through the mice caudal vessels for four weeks starting at 15 weeks.

After the behavioral tests, intraperitoneal glucose tolerance test (IPGTT) was conducted in all of the groups. Before the IPGTT, the mice were fasted for 12 h, and then D-glucose (2 g/kg of BW) was intraperitoneally injected. Blood glucose was measured in the tail vein blood at 0, 15, 30, 60, 90, and 120 min. from the injection point [18].

### 2.6. Behavioral Experiments in HFD-Induced Obese Mice

To estimate the novelty preference and memory without learning rules, a Y-maze test was conducted based on the mice’s innate tendency to explore a novel environment. The apparatus that was used in the experiment was a white plastic maze consisting of three arms (33 cm long, 15 cm high, and 10 cm wide) that were intersected at 120°. The mouse was placed at the end of one arm. After that, the mouse was allowed to move freely for 8 min., and spontaneous spatial preference behavior was recorded by the Smart 3.0 video tracking system (Panlab, Barcelona, Spain). The percentage of spontaneous spatial preference behavior was calculated according to the following formula: (Number of alternations/(Total number of arm entries-2))×100 [19].

A passive avoidance test was performed after the Y-maze test to assess short-term memory ability through training. The instrument that was used in the experiment was a plastic chamber with two spaces being separated by one gate. One space was bright, while the other space was dark, and it had a metal grid floor equipped with electricity flow. The experiment was carried out over two days. On the first day, the mouse was left in the bright space for 1 min., and when the gate was opened and the mouse completely passed into the dark side, the gate was closed and an electric shock (0.5 mA, 3 s) was applied on the foot. After the training trials, on the second day, the mouse was placed in the bright space and the step-through latency time that was taken to move to the dark space was measured (the step-through latency maximum testing time: 300 s) [19].

A Morris water maze test was conducted for six days to estimate the long-term memory and learning abilities of mice with long-term learning that relies on visual signs to navigate the platform from a starting point. For the test, a stainless circular pool (90 cm in diameter) was divided into four zones (E, W. S, and N) and there were visual signs for navigation on each wall. The pool was filled with water (22 ± 2 °C) with white milk powder being dissolved in it and a platform (6 cm in diameter) was placed in the middle of the N zone of Section 4 and its position did not change during the experiment. Training was performed for five days where the mice were placed at different starting points (E, W, S, and N) to recognize the platform location and the latency time of the mice was measured while using the Smart 3.0 video tracking system. On the last day, the travel trajectory of the mice was recorded by the Smart 3.0 video tracking system after removing the platform and placing the mice in the pool for 60 s [20].

### 2.7. Blood Serum Biochemical Analysis

Blood samples that were obtained at the same time as the dissection were immediately centrifuged at 15,000 rpm for 15 min. at 4 °C to acquire the serum. Concentration of the glutamic oxaloacetic transaminase (GOT), glutamic pyruvic transaminase (GPT), blood urea nitrogen (BUN), creatine (CRE), lactate dehydrogenase (LDH), total cholesterol (TCHO), triglyceride (TG), and high-density lipoprotein cholesterol (HDLC) were measured in the serum as clinical chemistry analyzer (Fuji Dri-chem 4000i; Fuji Film Co., Tokyo, Japan). Using the “Friedewald formula”, the LDLC concentration was calculated: LDLC (mg/dL) = TCHO − (HDLC + TG/5). The ratio of HDLC to TCHO is indicated by HTR (%), and the HTR is calculated as follows: HTR(%) = (HDLC/TCHO) × 100 [18].

### 2.8. Measurements of Antioxidant System and Cholinergic System from Brain Tissue

The following series of biochemical indicators were measured to confirm the effect of EFAD on HFD-induced oxidative stress and damaged cholinergic system. After the behavioral tests, the mice were sacrificed by CO_2_ inhalation. Brain tissue from the mice was immediately stored on ice and then minced and homogenized with cold phosphate buffered saline (PBS). After that, the brain tissue was kept at −80 °C until *ex-vivo* experiments were carried out [14].

### 2.9. Measurement of Antioxidant System

The pretreated brain tissue was centrifuged at 12,000× g for 30 min. at 4 °C. The supernatant was mixed with an alkaline hydroxylamine reagent (2 M hydroxylamine in HCl and 3.5 N sodium hydroxide) to measure the acetylcholine (Ach) levels. After 1 min., 0.5 N HCl and 0.37 M FeCl_3_ in 0.1 N HCl were added to the mixture and then absorbance was checked at 540 nm. 50 mM sodium phosphate buffer (pH 8.0) was added to the supernatant and then incubated at 37 °C for 5 min. to test AChE activity. After that, Ellman’s reaction mixture (1 mM DTNB and 0.5 mM ACh in 50 mM sodium phosphate buffer) was added. Absorbance was measured at 405 nm at intervals of 1 min. for 10 min. [21].

Supernatant was prepared by centrifuging pretreated brain tissue at 2500× g for 10 min. to measure the contents of malonaldehyde (MDA). 1% phosphoric acid and a 0.67% TBA solution were added to the supernatant and placed in a water bath (95 °C) for 1 h, and absorbance was then measured at 532 nm.

In the SOD test, pretreated brain tissue was centrifuged at 400× g for 10 min. for measuring the total SOD contents. The pellet was used for the SOD test according to the instructions of the manufacture of the SOD kit.

Brain tissue was homogenized with phosphate buffer and centrifuged at 10,000× g for 15 min. to measure reduced glutathione (GSH) levels. The supernatant was mixed with 5% metaphosphoric acid and centrifuged at 2000× g for 2 min. Tris-HCl buffer (0.26 M, pH 7.8), 0.65 N NaOH, and *O*-phthalaldehyde (OPT, 1 mg/mL) were added to the supernatant and left in the dark for 15 min. After that, it was measured with a fluorescent microplate reader (Infinite 200, Tecan Co., San Jose, CA, USA) (λ_excitation_: 320 nm, λ_emission_: 420 nm) [22].

### 2.10. Mitochondrial Activity Tests

For the mitochondrial experiments of the brain, experiments were carried out using five mice that were raised under the same condition as *in-vivo*. To separately isolate mitochondria, the procedures of Dragicevic were carried out. After that, the remaining pellet was used for other experiments [23].

Isolation buffer was added to the remaining pellet to estimate mitochondrial membrane potential (MMP), the reactive oxygen species (ROS), and ATP measurement (215 mM mannitol, 75 mM sucrose, 0.1% BSA, 20 mM HEPES sodium, and pH 7.2).

In the MMP assay, an assay buffer (EGTA-free buffer with 5 mM pyruvate and 5 mM malate) and 1 μM JC-1 were added to the previously preprocessed sample and then measured with a fluorescent microplate reader (λ_excitation_: 530/25 nm, λ_emission_: 590/30 nm) [14].

In the ROS assay, 25 μM 2′,7′-dichlorofluorescein diacetate was added to the previously preprocessed sample and the ROS levels were then quantified with a fluorescent microplate reader (λ_excitation_: 485/20 nm, λ_emission_: 528/20 nm) [14].

In the ATP assay, the previously preprocessed sample was centrifuged at 13,000× g for 10 min. The pellet was used for the ATP test according to the instructions of the manufacture of the ATP kit. ATP standard quantification curves were used to convert the ATP contents.

### 2.11. Western Blot Analysis

Liver, white adipose, and brain tissue were homogenized with ProtinExAnimal cell/tissue (GeneAll Biotechnology, Seoul, Korea) with 1% protease inhibitor cocktails (Thermo Fisher Scientific, Rockford, IL, USA). The lysates were separated by sodium dodecyl sulfate polyacrylamide gel electrophoresis and then transferred to a polyvinylidene difluoride membrane (Millipore, Billerica, MA, USA). The membranes were reacted with primary antibodies that were diluted with 1:1000 for overnight. Subsequently, the secondary antibody solutions were allowed to react with the membrane for 1 h. Finally, the membranes were examined by exposing the membranes to an ECL reagent (Bionote, Hwaseong, Korea). The luminescence was detected with a ChemiDoc system (iBrightTM CL1000 instrument, Invitrogen, Carlsbad, CA, USA) [17].

### 2.12. Statistical Analysis

All of the values were indicated as mean ± SD. All data were analyzed using Duncan’s new multiple-range test (*p* < 0.05) of SAS ver. 9.1 (SAS Institute Inc., Cary, NC, USA), and the statistical differences between each group were calculated by one-way analysis (ANOVA).

## 3. Results

### 3.1. In-Vitro Experiments on Obesity and Insulin Resistance in 3T3-L1 Cells

To evaluate the ability of EFAD to inhibit lipid accumulation, the lipids that were produced through 3T3-L1 cell differentiation were stained with Oil Red O, and the results are shown in Figure 1A. The cytotoxicity was confirmed in 200 μg/mL of EFAD, accordingly the experiment proceeded to a maximum concentration 100 μg/mL. The degree of lipid that was produced in MDI medium, which is optimized for lipid accumulation, was converted to 100% (control (MDI), 100.00 ± 2.28). As a result of treatment of 3T3-L1 cells with EFAD, it was confirmed that the higher the concentration of EFAD, the more the lipid accumulation is inhibited (EFAD10, 95.20 ± 5.22; EFAD20, 90.58 ± 3.36; EFAD50, 86.89 ± 4.25; EFAD100, 75.10 ± 0.95%, respectively).

Figure 1B shows the effect of EFAD on 3T3-L1 cells in which IR was induced by TNF-α treatment. The degree of p-AMPK was significantly decreased in the TNF-α group (0.77 ± 0.02), and the corresponding value in the control group (1.03 ± 1.00) was confirmed in the EFAD groups (EFAD50, 1.07 ± 0.12; EFAD100, 1.08 ± 0.10, relatively). In addition, the degree of p-IRS(ser307) was significantly increased in the TNF-α group (relative density: 1.00 ± 0.04) and decreased in the EFAD groups (EFAD50, 0.90 ± 0.16; EFAD100, 0.77 ± 0.10, relatively), and, in particular, the EFAD50 group showed a similar degree as the control group (0.80 ± 0.02). Finally, p-Akt was significantly reduced in the TNF-α group (0.75 ± 0.07). On the other hand, in the EFAD groups (EFAD50, 0.99 ± 0.02; EFAD100, 1.03 ± 0.10, relatively), values that corresponded to the control group (0.96 ± 0.02) were shown.

### 3.2. In-Vivo Experiments on Obesity and Impaired Glucose Tolerance in C57BL/6 Mice

To estimate the anti-obesity effect of EFAD, the averages of body weight (BW) of each group were measured in HFD-induced-obese mice. There was no significant difference in the initial BW of all groups, and the increase of BW in the EFAD groups were slower than in the HFD group. However, the food intake was not significantly different between the HFD and EFAD groups (Figure 2A, Appendix A). To confirm the effect of EFAD on diabetic symptoms that are caused by obesity, the results of fasting blood glucose at 15 weeks after the HFD and for four weeks after starting the EFAD diet are shown in Figure 2B. In the control group, the fasting blood glucose level was normal at 15 weeks (93.71 ± 15.94 mg/dL), whereas the HFD groups had a high fasting blood glucose level of 160–170 mg/dL at 15 weeks (HFD, 167.57 ± 14.64; EFAD20, 169.57 ± 9.80; EFAD40, 171.43 ± 15.86 mg/dL, respectively). The EFAD diet had a significant effect in reducing the fasting blood glucose levels at 19 weeks (control, 120.71 ± 13.00; HFD, 174.71 ± 8.73; EFAD20, 150.29 ± 8.71; EFAD40, 153.00 ± 5.16 mg/dL, respectively).

The results of an intraperitoneal glucose tolerance test (IPGTT) (Figure 2C) to evaluate glucose tolerance showed that the blood glucose level rapidly rose after the glucose injection in the HFD group and that it took a long time to recover. In contrast, the EFAD40 group showed a tendency to recover the impaired glucose tolerance caused by HFD. The area under the curve (AUC) with IPGTT showed that the HFD group (4.94 ± 0.69 × 104 mg/dL*min) had a much higher value than the control group (2.26 ± 0.47 × 104 mg/dL*min), and the AUC value decreased in the EFAD groups (EFAD20, 4.66 ± 0.39; EFAD40, 4.05 ± 0.48 × 104 mg/dL*min, respectively) (Figure 2D).

### 3.3. In-Vivo Behavioral Analysis Tests in C57BL/6 Mice

Figure 3 shows the results of experiments that were performed to observe the improvement effect of EFAD on cognitive impairment caused by HFD. In a Y-maze test (Figure 3A–C), there was no difference in exercise capacity among all of the groups (Figure 3B). The HFD group had very low alternation behavior (26.92 ± 6.74%), which confirmed that the innate spatial exploration ability is injured by the HFD. On the other hand, in the EFAD groups (EFAD20, 37.71 ± 8.29; EFAD40, 44.28 ± 10.83%, respectively), it was confirmed that the spatial perception ability that was damaged by the HFD was improved (Figure 3A). As a result of converting the mice movement path into three-dimensional (3D) data in the Y-maze test for 8 min. (Figure 3C), the HFD group showed an intensive distribution in one place. By contrast, even distribution in each path was confirmed for the control group and the EFAD groups.

A passive avoidance test, which is used to estimate short-term memory that formed by the fear of electric shock, was conducted, and Figure 3D shows the results. It showed that the step-through latency in the HFD group was reduced (95.00 ± 37.18 s), and those of the EFAD groups (EFAD20, 179.40 ± 112.80; EFAD40, 281.80 ± 40.70 s, respectively) were remarkably improved.

To examine the long-term memory ability that formed by repetitive learning for four days, a Morris water maze test was performed, and the results are shown in Figure 3E–G. Each group of mice noticed the presence of a hidden platform by learning for four days, but it was confirmed that the HFD group took a long time to reach the platform, despite repeated learning (Figure 3E). After removing the platform, the measurement of the time of staying in the N zone where the platform existed showed that the HFD group (25.62 ± 9.11%) had impaired cognitive ability. In contrast, in the EFAD groups (EFAD20, 37.69 ± 4.41; EFAD40, 42.10 ± 12.91%, respectively) (Figure 3F), it was confirmed that the damaged long-term memory ability caused by HFD was improved. In particular, the path trace data conducted for 1 min. in each group showed damaged spatial and learning ability in the HFD group. In contrast, the EFAD groups recognized where the platform had been located (Figure 3G).

### 3.4. Ex-Vivo Experiment on Blood Serum Analysis

Table 1 shows the results of serum analysis to identify liver damage, renal damage, and dyslipidemia. Blood serum analysis showed that the levels of GOT and GPT, which are liver injury factors, were very high in the HFD group when compared to the control group. In contrast, EFAD effectively lowered the liver damage levels. BUN and CRE were measured for the evaluation of renal damage, and there was no significant difference in the results. In addition, the LDH measurement for cytotoxicity evaluation showed a decrease in the levels in the EFAD groups. The results of TCHO, TG, and low-density lipoprotein cholesterol (LDL, mg/dL) levels, and HTR showed that dyslipidemia was induced in the HFD group when compared with the control group. The EFAD20 group and EFAD40 group were effectively improved.

### 3.5. Ex-Vivo Experiments on Cholinergic System in Brain Tissue

The results of an experiment to identify the impaired cholinergic system, which is one of the pathological features of AD, are shown in Figure 4. In the HFD group, decreased ACh contents (0.42 ± 0.10 mmol/mg of protein) and increased AChE activity (112.81 ± 5.94%) and AChE expression levels (relative density: 1.00 ± 0.13) were confirmed. In contrast, the ACh contents (EFAD20, 0.52 ± 0.12; EFAD40, 0.54 ± 0.08 mmol/mg of protein, respectively) increased and the AChE activity (EFAD20, 112.16 ± 5.34; EFAD40, 107.14 ± 6.70%, respectively) and AChE expression levels (relative density: EFAD20, 0.78 ± 0.07; EFAD40, 0.72 ± 0.05, respectively) statistically decreased in the EFAD groups.

### 3.6. Ex-Vivo Experiments on Antioxidant System in Brain Tissue

The MDA levels were measured in brain tissue as a bio-indicator measuring lipid peroxidation. Figure 5A shoes the results. High MDA contents were confirmed in the HFD group (12.96 ± 2.50 nmol/mg of protein). The MDA contents of the EFAD groups (EFAD20, 9.40 ± 1.80; EFAD40, 8.65 ± 2.11 nmol/mg of protein, respectively) were statistically significant with the control group (9.40 ± 0.17 nmol/mg of protein).

As a result of confirming the SOD and GSH levels, which play a major role in the antioxidant system in the body (Figure 5B,C), SOD and reduced GSH contents in the HFD group (SOD, 2.66 ± 0.87 nmol/mg of protein; GSH, 1.78 ± 0.07 μg GSH/mg of protein) were drastically reduced, which means that the antioxidant system was damaged in the HFD group. EFAD effectively restored the damaged antioxidant system by increasing the contents of SOD (EFAD20, 3.55 ± 0.75; EFAD40, 4.26 ± 0.64 nmol/mg of protein) and GSH (EFAD20, 1.93 ± 0.04; EFAD40, 2.18 ± 0.16 μg GSH/mg of protein) in the EFAD groups.

### 3.7. Ex-Vivo Experiments on Mitochondrial Activity in Brain Tissue

As a result of measurement of MMP (Figure 6A), which plays a major role in mitochondrial activity, MMP decreased in the HFD group (75.87 ± 4.15%), and MMP increased in the EFAD groups (EFAD20, 84.73 ± 4.53; EFAD40, 89.85 ± 4.67%, respectively). Figure 6B shows the results of confirming ROS levels in the mitochondria. High ROS levels were observed in the HFD group (109.68 ± 2.34%) and they were lower in the EFAD groups (EFAD20, 104.26 ± 0.60; EFAD40, 103.39 ± 0.50%, respectively). In addition, the ATP levels in the mitochondria were examined (Figure 6C) and it was confirmed that mitochondrial energy biosynthesis was impaired by the low ATP level in the HFD group (39.52 ± 29.63 pmol/mg of protein) and the ATP biosynthesis capacity of mitochondria was improved in the EFAD groups (EFAD20, 79.99 ± 22.18; EFAD40, 137.16 ± 33.48 pmol/mg of protein, respectively).

### 3.8. Ex-Vivo Experiments on Peripheral-Insulin Resistance-Related Factors in White Adipose and Liver Tissue

Figure 7 shows the expression levels of p-IRS(ser307) and TNF-α in peripheral tissues, such as white adipose tissue (WAT) and liver. The expression of TNF-α was significantly increased in the HFD group, confirming that chronic inflammation was induced in peripheral tissues (relative density: WAT, 0.31 ± 0.06; liver tissue, 0.81 ± 0.15, respectively). Thus, EFAD40 was shown to effectively improve chronic inflammation (WAT, 0.22 ± 0.07; liver tissue, 0.69 ± 0.02, respectively). The expression of p-IRS(ser307) was greatly increased in the HFD group (WAT, 0.37 ± 0.06; liver tissue, 0.81 ± 0.23, respectively). In contrast, it was confirmed that EFAD40 decreased the expression of p-IRS(ser307) (WAT, 0.27 ± 0.01; liver tissue, 0.56 ± 0.14, respectively).

### 3.9. Ex-Vivo Experiments on CNS-Insulin Resistance-Related Factors in Brain Tissue

Figure 8A shows the IRS/Akt pathway results. In the HFD group, p-Akt reduction (relative density: 0.49 ± 0.17) was confirmed along with p-IRS(ser307) (0.68 ± 0.02) and p-tau(ser404) (0.77 ± 0.03) increase. In contrast, in the EFAD groups, p-IRS(ser307) decreased (EFAD20, 0.54 ± 0.07; EFAD40, 0.44 ± 0.05, respectively) and p-Akt increased (EFAD20, 0.54 ± 0.18; EFAD40, 0.79 ± 0.02, respectively). In addition, the phosphorylation of tau decreased (EFAD20, 0.67 ± 0.04; EFAD40, 0.50 ± 0.07, respectively). Figure 8B shows the results of the AMPK/mTOR pathway. In the HFD group, p-AMPK decreased (0.43 ± 0.09) and IDE level decreased (0.71 ± 0.06). Additionally, it was confirmed that the amount of Aβ increased (0.35 ± 0.06). In the other hand, in the EFAD groups, AMPK was activated (EFAD20, 0.70 ± 0.09; EFAD40, 0.84 ± 0.09, respectively) and the IDE level increased (EFAD20, 0.77 ± 0.06; EFAD40, 0.84 ± 0.10, respectively). Therefore, it was confirmed that the Aβ level decreased (EFAD20, 0.30 ± 0.06; EFAD40, 0.28 ± 0.05, respectively).

## 4. Discussion

Through various studies, various physiological activities of *Aruncus dioicus* var. *kamtschaticus* in have been proved [11,12,13,14]. In animal experiments, it has been confirmed that the *Aruncus dioicus* var. *kamtschaticus* is effective for improving diabetes and cognitive function [11,14]. In this study, we have focused on the improvement effect of EFAD on cognitive impairment by implementing AD characteristics according to peripheral- and CNS-IR.

Obesity causes various complications by inducing chronic low-grade inflammation due to adipose tissue enlargement as well as metabolic dysfunction that are caused by chronic over-nutrition [2,4,5,24]. These persistent metabolic dysfunction and inflammatory responses further impair insulin sensitivity in cells, which leads to disorders of glucose metabolism and eventually to T2DM [2,3]. Recent studies suggest that IR in peripheral tissues in T2DM patients correlates with IR in the CNS, thereby linking AD [6,7].

The enlargement and hypertrophy of adipocyte, as an obesity characteristic, cause macrophage infiltration into adipose tissue due to cytokines that are derived from adipocytes and immune responses. Additionally, the expansion of adipose tissue induces NF-κB transcription of macrophages due to the excessive release of free fatty acids in adipose tissue, resulting in the release of cytokines, such as TNF-α [2,3,4]. This chronic low-grade inflammation is known to induce peripheral-IR. Thus, in this experiment, the effects of EFAD on obesity and IR caused by inflammation reaction were confirmed by the 3T3-L1 experiments (Figure 1). The results of the *in-vitro* experiments showed that EFAD effectively inhibited lipid accumulation in Oil Red O staining, and improved IR in pro-inflammation induced 3T3-L1 cells. In the insulin signaling process of normal cells, when insulin binds to the insulin receptor, the tyrosine residues of IRS are phosphorylated, and then the glucose transporter (GLUT) that is present in the cytoplasm is translocated to the cell membrane to absorb glucose from the blood into the cell. However, in the case of IR-induced cells, the downstream is disturbed by the serine residue phosphorylation of IRS as an inactive state, and glucose absorption is not achieved [7,15,16,24]. AMPK is a major factor that is involved in energy homeostasis, and it plays a role in regulating various cellular metabolic pathways, such as intracellular glucose uptake and fatty acid and protein synthesis, and the oxidation of fatty acid [25]. In particular, the phosphorylation of AMPK has been shown to increase the translocation of GLUT4 to the cell membrane in the IR cells [15,26]. We confirmed that IR eventually occurs in TNF-α treated 3T3-L1 cells by increasing p-IRS(ser307) and reducing p-Akt. EFAD effectively improved IR in the TNF-α treated 3T3-L1 cells. Additionally, p-AMPK is decreased in TNF-α treated 3T3-L1 cells, while EFAD increased p-AMPK. Therefore, in the IR that was caused by the inflammatory reaction, EFAD activates IRS, Akt, and AMPK, which in turn may have a potential effect on IR cells by improving energy homeostasis.

The body lives by producing energy through normal metabolic and nutritional processes. However, entry into a chronic over-nutrition state leads to the overproduction of ROS along with excessively generated energy [24]. The ROS break down the redox balance *in-vivo* and this eventually leads to mitochondrial dysfunction, and the excess energy accumulates as intracellular lipid. Furthermore, chronic low-grade inflammation caused by obesity, as well as ROS, leads to reduced insulin sensitivity of the cells, resulting in IR [24,25]. In this study, we showed that HFD can cause fat accumulation in the body and result in impaired glucose tolerance as metabolic imbalance symptoms. Additionally, we confirmed that EFAD could effectively reduce BW gain in HFD-induced-obese mice and improve the impaired glucose tolerance that is caused by HFD (Figure 2A,B). According to a study by Rains et al., the phenolics present in foods are known to have anti-obesity effects by promoting energy consumption and lipid oxidation, inhibiting appetite, delaying nutrient absorption, and alternating in fat metabolism [10]. EFAD seems to have influenced the enhancement of energy homeostasis and the inhibition of lipid accumulation, as confirmed by 3T3-L1 cell experiments. That is, these results suggest that EFAD has anti-obesity and anti-diabetes effects by improving energy homeostasis by inhibiting lipogenesis and promoting fatty acid oxidation in adipocytes. Furthermore, previous studies have shown that EFAD has an excellent antioxidant effect, along with an inhibitory effect on mitochondrial mediated-apoptosis and α-glucosidase [13,14]. Based on the effects of EFAD, phenolics, such as the dicaffeoyl glucose isomers in EFAD, not only improve mitochondrial metabolism, but also regulate energy homeostasis, and thus have potential for anti-obesity and anti-diabetic effects.

Several studies have suggested that the peripheral-IR ultimately leads to CNS-IR. Thus, some scholars believe that the link between T2DM and AD is IR, and AD is referred to as T3DM [6,7,26]. In this study, behavioral tests of HFD-induced obese and/or diabetic-mice were performed and the results showed cognitive impairment in the HFD group. On the other hand, the EFAD groups effectively improved cognitive dysfunction (Figure 3). In the case of peripheral IR, the β-pancreatic cells recognize insulin deficiency and continuously increase insulin secretion, resulting in hyperinsulinemia. Insulin reaches the CNS across the blood-brain barrier (BBB) through a receptor-mediated process. Chronic hyperinsulinemia eventually triggers CNS-IR, thereby reducing the insulin levels in the brain via the BBB, affecting energy homeostasis and cognitive disruption [25]. Baskin’s studies have shown that direct injection of insulin into the brain enhances the cognition and memory ability in mice, which emphasizes the importance of insulin in the CNS, but the role of insulin in the CNS has not yet been established [27]. We suggest that peripheral-IR induced by HFD may damage the CNS and affect cognitive impairment, based on *in-vitro* and *in-vivo* experiments. We investigated the effects of EFAD on peripheral IR and CNS IR by identifying the major factors in peripheral tissues and brain tissues and identifying AD characteristics in subsequent experiments.

The expression of p-IRS(ser307) and TNF-α in WAT and liver tissue were measured (Figure 7), and dyslipidemia, a major characteristic of peripheral-IR, was confirmed by blood serum biochemical analysis to determine the effect of EFAD on peripheral tissue (Table 1). It was ascertained that chronic low-grade inflammation were induced, and furthermore, IR was induced in peripheral tissues by confirming the increased levels of inflammation in peripheral tissues caused by HFD. Additionally, dyslipidemia, which showed increased total cholesterol level and decreased HTR in the HFD group, was confirmed. In contrast, it was confirmed that EFAD excellently improved peripheral-IR. Lipids and cholesterol constitute an important part of the brain to maintain a turnover in the synapse and cell connections of the CNS. Thus, dyslipidemia that is caused by peripheral-IR has a major adverse effect on the CNS [25]. In addition, according to previous research [28], hepatic insulin resistance, oxidative stress, and injury together promote the increased generation of “toxic lipids”, such as ceramides, which can cross the BBB; they reach the brain, and thereby exert neurodegenerative effects *via* a liver-brain axis. The GOT and GTP as the liver injury factors in the EFAD group were also decreased when compared with the HFD group. Eventually, these results suggest that EFAD may be effective in improving cognitive impairment due to HFD.

We confirmed the effects of EFAD by identifying the cholinergic system, mitochondrial dysfunction, oxidative stress, and neurotoxic substances in AD associated with CNS-IR to demonstrate the effect of EFAD. 

ACh is a neurotransmitter that is synthesized by choline actyltransferase and hydrolyzed by AChE and it plays a major role in learning and memory in the brain. AD is known to exert excessive activation of AChE and thereby inhibit the role of cholinergic neurotransmitters, thus affecting cognition, and furthermore affecting neuronal apoptosis [29]. IR mainly breaks down energy homeostasis by mitochondrial dysfunction and impairs the role of insulin, which is known to regulate the neurotransmitters ACh and norepinephrine, affecting cognition [25]. Thus, the mitochondrial and neurotransmitter damage that are caused by CNS IR also lead to radical oxidative stress of the CNS, which interferes with neurotransmission, resulting in cognitive impairment. Previous *in-vitro* experiments have shown that EFAD has an excellent AChE inhibitory effect [13], and Schwaiger’s study [30] reported that the dicaffeoyl glucose of *Aruncus dioicus* has the best AChE inhibitory activity. That is, EFAD has been shown to improve cholinergic system impairment that is caused by HFD-induced CNS-IR through the excellent AChE inhibitory effect of dicaffeoyl glucose. 

AD is a neurodegenerative disease that is directly associated with metabolic disturbances due to aging-induced mitochondrial dysfunction [6]. CNS-IR that is caused by peripheral-IR induces insulin deficiency and hyperglycemia in the brain, resulting in damage to mitochondrial metabolism and neuronal apoptosis, and further causing excessive levels of oxidative stress [7,23,25]. Oxidative stress is known to be involved in causing diseases, such as non-alcoholic fatty liver disease (NAFLD) and AD. The precise mechanism of the onset and progression of NAFLD remains unclear, but it is suggested that oxidative stress may play a fundamental role [31]. In addition, HFD induced oxidative stress is associated obesity and AD markers besides insulin resistance, according to report of Nuzzo et al. [32].

The present study confirmed that HFD caused impaired mitochondrial activity and increased the oxidative stress in the brain, and EFAD effectively improved it (Figure 5 and Figure 6). A previous study has reported that EFAD inhibits neuron apoptosis on the basis of superior antioxidant effects in the central mitochondrial mediated-apoptosis pathway [14]. Thus, the excellent antioxidant and peripheral-IR improvement effects of EFAD may contribute to the improvement of CNS-IR.

The most prominent feature of AD’s pathological features is Aβ plaque formation and tangled nerve fibers [7]. The serine residue of IRS is phosphorylated when IR is induced in the CNS, thereby decreasing the activity of Akt, resulting in the hyperphosphorylation of tau protein. Hyperphosphorylated tau induces nerve fiber entanglement in neurons, which interferes with neurotransmission and acts as a toxic substance, inducing apoptosis [25]. In addition, the activity of AMPK is reduced due to IR, which causes an imbalance in energy homeostasis. Recently, AMPK is reported to be involved in IDE, which degrades Aβ as well as insulin secretion through a non-traditional secretory pathway, known as the AMPK/mTOR pathway [33]. In other words, the inactivation of AMPK that is induced by IR reduces IDE secretion from astrocytes and eventually decreases the degradation of neuronal extracellular Aβ plaques, which in turn promotes plaque formation. Thus, in this experiment, western blotting confirmed hyperphosphorylated tau and Aβ, and high levels in the HFD group suggested that HFD-induced IR eventually contributed to the neurotoxicity of the CNS (Figure 8). In contrast, EFAD had an outstanding inhibitory effect on the production of neurotoxicants in the CNS.

In a previous study, it was confirmed that the major compound contained in EFAD is dicaffeoyl glucose isomers as 3,4-dicaffeoyl-α-D-glucoside and 3,4-dicaffeoyl-β-D-glucoside through UPLC-Q-TOF/MS^2^ [14]. These substances contribute to the excellent effect of EFAD on mitochondrial damage and oxidative stress. Additionally, the caffeic acid in *Aruncus dioicus* is a major substance that was confirmed by HPLC in Park’s study [13], and it has an excellent α-glucosidase inhibitory effect. It is reported that caffeic acid with a large amount of *O*-dihydroxy group that is attached to the phenolic ring continuously increases the antioxidant activity [34]. Therefore, it is suggested that a caffeic acid-inducing compound, which is a main substance of EFAD, is involved in energy metabolism based on excellent antioxidant power as well as α-glucosidase inhibitory effect, thereby further suppressing various symptoms that are caused by metabolic imbalance.

In summary, we demonstrated the *in-vitro* anti-obesity and IR improvement effect of EFAD while using the 3T3-L1 cell. After that, EFAD was found to be effective in reducing weight and improving glucose tolerance and cognitive impairments in obese and diabetic mice induced by HFD with *in-vivo* tests. In addition, serum and peripheral tissue analysis showed that EFAD was effective on HFD-induced peripheral IR. Furthermore, we confirmed the interrelated symptoms of CNS-IR and AD in the brain, such as oxidative stress, cholinergic system, mitochondrial damage, and neurotoxin production. As a result, EFAD improved the pathological characteristics of HFD-induced neurodegeneration, such as AD. The effects of EFAD are believed to have been caused by the dicaffeoyl glucose isomers. Therefore, further studies on this substance are considered to be necessary. Consequently, phenolics, such as the dicaffeoyl glucose isomers contained in EFAD, are expected to have an excellent effect on final cognitive impairment by improving the symptoms that are caused by metabolic imbalance. Therefore, this study suggests the possibility of using *Aruncus dioicus* var. *kamtschaticus* as a functional food substance for the improvement of cognitive dysfunction that is caused by metabolic imbalance by studying the relationship between IR and neurodegeneration caused by HFD.

## 5. Conclusions

In summary, we demonstrated the in-vitro anti-obesity and IR improvement effect of EFAD while using the 3T3-L1 cell. After that, EFAD was found to be effective in reducing weight and improving glucose tolerance, and cognitive impairments in obese and diabetic mice induced by HFD with in-vivo tests. In addition, serum and peripheral tissue analysis showed that EFAD was effective on HFD-induced peripheral IR. Furthermore, we confirmed the interrelated symptoms of CNS-IR and AD in the brain, such as oxidative stress, cholinergic system, mitochondrial damage, and neurotoxin production. As a result, EFAD improved the pathological characteristics of HFD-induced neurodegeneration, such as AD. The dicaffeoyl glucose isomers are believed to have caused the effects of EFAD. Therefore, further studies on this substance are considered to be necessary. Consequently, phenolics, such as the dicaffeoyl glucose isomers contained in EFAD, are expected to have an excellent effect on final cognitive impairment by improving the symptoms that are caused by metabolic imbalance. Therefore, this study suggests the possibility of using *Aruncus dioicus* var. *kamtschaticus* as a functional food substance for the improvement of cognitive dysfunction that is caused by metabolic imbalance by studying the relationship between IR and neurodegeneration caused by HFD.

## Figures and Tables

**Figure 1 nutrients-11-01319-f001:**
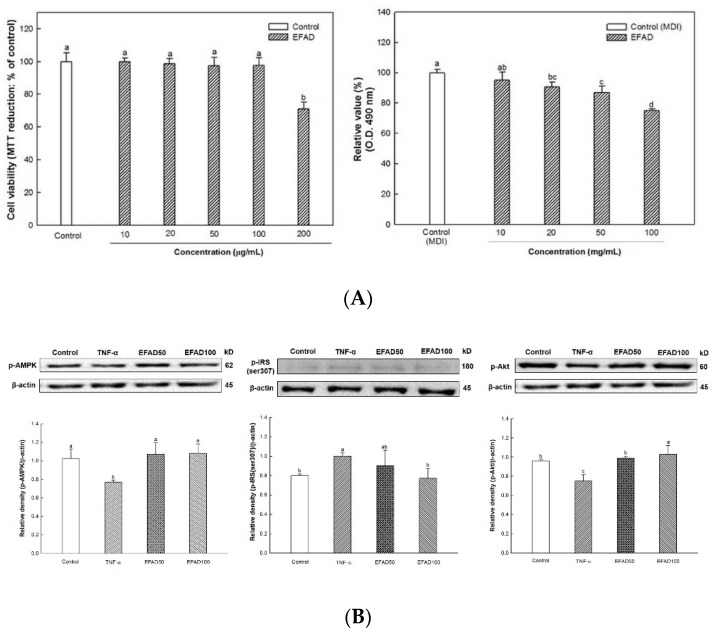
Effects of ethyl acetate fraction from *Aruncus dioicus* var. *kamtschaticus* (EFAD) against cytotoxicity, lipid accumulation, and insulin resistance on 3T3-L1 cell. (**A**) Cell viability by MTT assay and degrees of lipid accumulation by Oil Red O staining method. (**B**) Expression degrees of phosphorylated-AMP activated kinase (p-AMPK), phosphorylated-insulin receptor substrate (p-IRS) (ser307), and phosphorylated-protein kinase B (p-Akt) on Tumor necrosis factor alpha (TNF-a). The values shown are as mean ± SD (*n* = 3; “*n*” means the technical repeats). The data were statistically considered at *p* < 0.05 and were represented as different small letters.

**Figure 2 nutrients-11-01319-f002:**
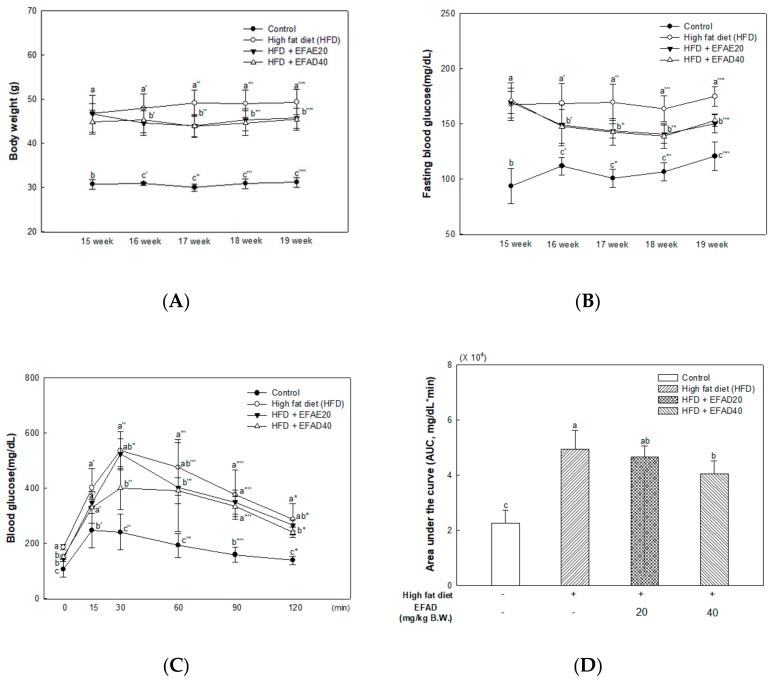
Effect of ethyl acetate fraction from *Aruncus dioicus* var. *kamtschaticus* (EFAD) on obesity and impaired glucose tolerance induced by high-fat diet (HFD). (**A**) Average of body weight. (**B**) Fasting blood glucose from 15 week to 19 week. (**C**) Intraperitoneal glucose tolerance (IPGTT) test. (**D**) Area under the curve (AUC) in IPGTT. The values shown are as mean ± SD (*n* = 7). The data were statistically considered at *p* < 0.05 and they were represented as different small letters.

**Figure 3 nutrients-11-01319-f003:**
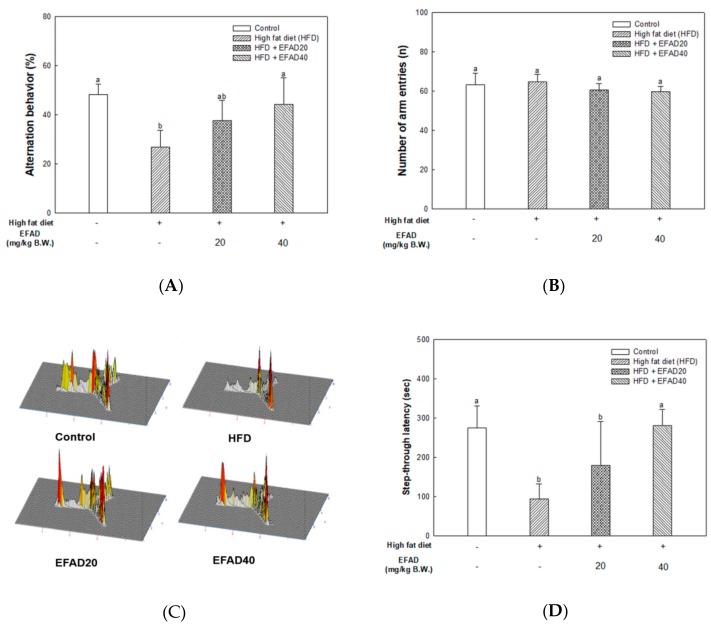
Effect of ethyl acetate fraction from *Aruncus dioicus* var. *kamtschaticus* (EFAD) on cognitive dysfunction induced by High fat diet (HFD). (**A**) Alternation behavior. (**B**) Number of arm entry. (**C**) Three-dimensional (3D) data of tracking path in Y-maze test. (**D**) Step-through latency time measured by passive avoidance test. (**E**) Escape latency time from one day to four day. (**F**) Retention time in N zone. (**G**) Travel trajectories measured in Morris water maze test. The values shown are as mean ± SD (*n* = 7). The data were statistically considered at *p* < 0.05 and they were represented as different small letters.

**Figure 4 nutrients-11-01319-f004:**
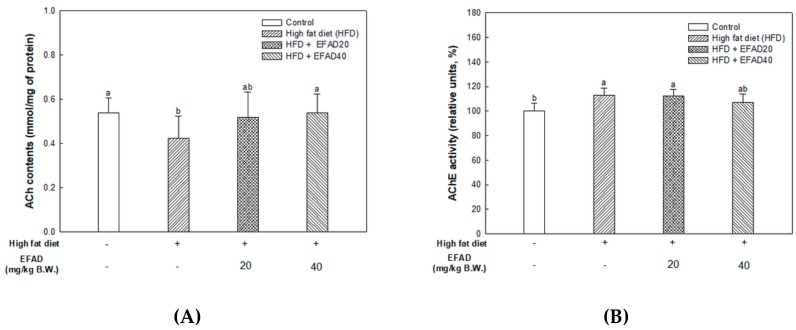
Effect of ethyl acetate fraction from *Aruncus dioicus* var. *kamtschaticus* (EFAD) on impaired cholinergic system induced by High fat diet (HFD) in brain tissue of mice. (**A**) Acetylcholine (ACh) contents. (**B**) Acetylcholinesterase (AChE) activity. (**C**) Expression degrees of AChE measured by western blotting analysis. The values shown are as mean ± SD ((**A**,**B**), *n* = 7; (**C**) *n* = 3). The data were statistically considered at *p* < 0.05 and were represented as different small letters.

**Figure 5 nutrients-11-01319-f005:**
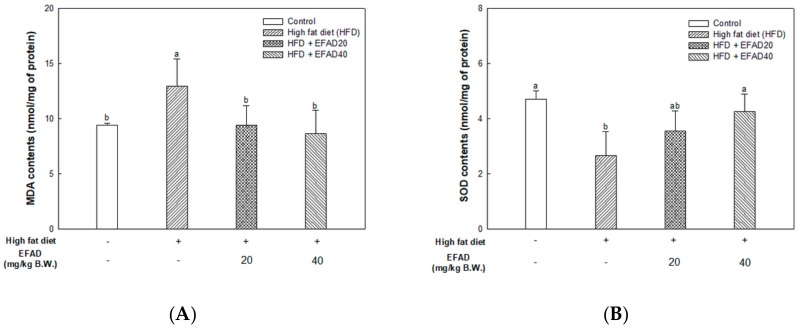
Effect of ethyl acetate fraction from *Aruncus dioicus* var. *kamtschaticus* (EFAD) on oxidative stress and impaired antioxidative system induced by HFD in brain tissue of mice. (**A**) Malonaldehyde (MDA) contents. (**B**) Superoxide dismutase (SOD) contents. (**C**) Reduced glutathione (GSH) contents. The values shown are as mean ± SD (*n* = 7). The data were statistically considered at *p* < 0.05 and they were represented as different small letters.

**Figure 6 nutrients-11-01319-f006:**
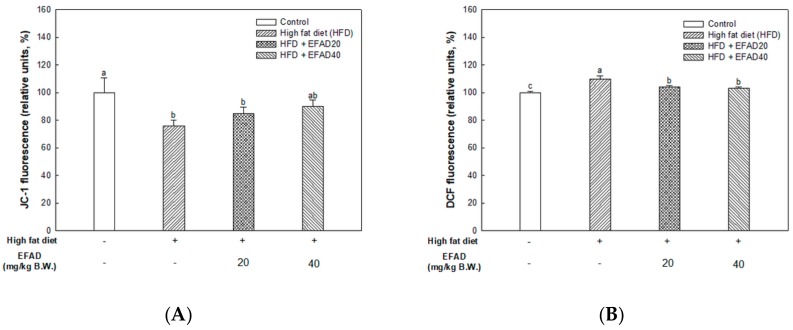
Effect of ethyl acetate fraction from *Aruncus dioicus* var. *kamtschaticus* (EFAD) on mitochondrial dysfunction induced by HFD in brain tissue of mice. (**A**) Mitochondrial membrane potential (MMP). (**B**) Degree of ROS levels in mitochondria. (**C**) Mitochondrial ATP levels. The values shown are as mean ± SD (*n* = 5). The data were statistically considered at *p* < 0.05 and were represented as different small letters.

**Figure 7 nutrients-11-01319-f007:**
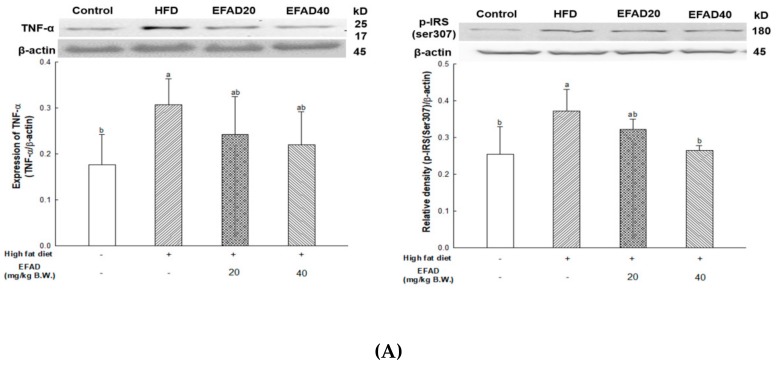
Effect of ethyl acetate fraction from *Aruncus dioicus* var. *kamtschaticus* (EFAD) against peripheral-insulin resistance induced by HFD. (**A**) Expression degrees of TNF-α and p-IRS(ser307) in white adipose tissue (WAT). (**B**) Expression degrees of TNF- and p-IRS(ser307) in liver tissue. The values shown are as mean ± SD (*n* = 3). The data were statistically considered at *p* < 0.05 and were represented as different small letters.

**Figure 8 nutrients-11-01319-f008:**
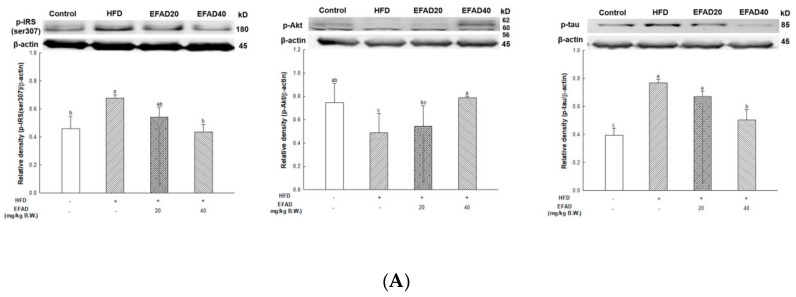
Effect of ethyl acetate fraction from *Aruncus dioicus* var. *kamtschaticus* (EFAD) against central nerve system-insulin resistance induced by HFD. (**A**) Expression degrees of p-IRS(ser307), p-Akt, and p-tau in brain tissue. (**B**) Expression degrees of p-AMPK, IDE, and amyloid β in brain tissue. The values shown are as mean ± SD (*n* = 3). The data were statistically considered at *p* < 0.05 and they were represented as different small letters.

**Table 1 nutrients-11-01319-t001:** Effect of ethyl acetate fraction from from *Aruncus dioicus* var. *kamtschaticus* (EFAD) on blood serum characteristics of High fat diet (HFD)-induced obese mice.

	Control	HFD	EFAD20	EFAD40
GOT (U/L)	40.20 ± 3.11 ^c^	108.00 ± 21.83 ^a^	82.40 ± 23.08 ^b^	75.80 ± 15.45 ^b^
GPT (U/L)	29.60 ± 4.28 ^b^	138.20 ± 38.51 ^a^	118.20 ± 58.96 _a_	85.20 ± 44.38 ^ab^
BUN (mg/dL)	17.36 ± 0.12 ^a^	16.68 ± 1.60 ^a^	15.52 ± 3.29 ^a^	17.72 ± 2.39 ^a^
CRE (mg/dL)	0.12 ± 0.04 ^a^	0.12 ± 0.04 ^a^	0.12 ± 0.04 _a_	0.10 ± 0.00 ^a^
LDH (mg/dL)	208.60 ± 52.18 ^c^	580.00 ± 176.27 ^a^	380.80 ± 157.68 _b_	308.60 ± 79.04 ^bc^
TCHO (mg/dL)	117.00 ± 4.74 ^c^	238.00 ± 27.34 ^a^	220.80 ± 26.02 ^ab^	208.20 ± 13.59 _b_
TG (mg/dL)	82.00 ± 15.31 _c_	160.20 ± 36.83 ^a^	125.00 ± 12.49 _b_	107.20 ± 28.12 ^bc^
HTR (%) *	62.86 ± 6.74 ^b^	59.79 ± 14.50 ^b^	65.82 ± 10.82 ^ab^	75.23 ± 8.63 ^a^
LDL (mg/dL) *	27.20 ± 6.15 ^b^	61.16 ± 23.71 ^a^	48.40 ± 14.25 ^ab^	29.96 ± 18.38 ^b^

* HTR (%) = high density lipoprotein cholesterol (HDLC)/TCHO × 100. * LDLC (mg/dL) = TCHO − (HDLC + TG/5). The results shown are as means ± SD (*n* = 7). The data were statistically considered at *p* < 0.05 when comparing with the control group and statistical differences were represented as different small letters.

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
