# Peer review of "Effect of Aruncus dioicus var. kamtschaticus Extract on Neurodegeneration Improvement: Ameliorating Role in Cognitive Disorder Caused by High-Fat Diet Induced Obesity"

_nutrients, 2019, doi:10.3390/nu11061319_

Round 1

Reviewer 1 Report

In the manuscript, the Authors investigated the neurodegenerative prevention role of ethyl acetate extract from Aruncus dioicus var. kamtschaticus (EFAD) by in vitro and in vivo tests. For in vitro experiments the Authors used EFAD extract in 3T3-L1 cells to prevent lipid accumulation and insulin resistance. The in vivo experiments were performed to demonstrate the EFAD anti-neurodegenerative role in HFD mice. Despite the interesting topic, the manuscript needs specific improvements.

1. During the in vitro tests the Authors used a concentration range of EFAD to demonstrate the reduction of lipid accumulation in 3T3-L1 cells. Could the Authors explain why they didn't use MTS/MTT assay to study cell viability as a function of time and concentration dependence after EFAD administration?

2. The Author described a cell increase of p-IRS at ser 307 residue. It is essential to demonstrate that the total insulin receptors remain unchanged. Therefore, I suggest to repeat this western blot whit anti-IRS.

3. It’s not completely clear the reason why the Authors use a cell model that is overcome by the experiments in mouse. In fact, in cells they do not investigate the specific pathways involved in reversion of insulin resistance after treatment whit EFAD extract.

4. In figure 1 the Authors used n=3. Please indicate what they mean for n. Are technical or biological repeats?

5. In figure 2(a) the Authors show the mouse body weight after 15 weeks but no information is given for early times. I suggest to start the lines in plot 2a with the initial average body weight of each group. In addition, there is no information provided about the food intake.

6. In table 1 are shown some markers concerning the liver damage. To correlate this condition to neurodegeneration, I suggest to add the reference Nuzzo D, et al., Nutrients 2018

7. In figure 6(a) JC-1 was used to measure the mitochondrial stress, but this probe is used in live mitochondria. Are the Authors work whit the live mitochondria? In Addition, in figure 6(b) was used Dichloro-dihydro-fluorescein diacetate (DCFH-DA) assay to estimate the mitochondria oxidative stress but this is not particular for this organelle. I suggest the more specific use of MitoSox.

8. At the end, the neuroprotective effect of EFAD extract requires a stronger evidence. I suggest to make a TUNEL assay to evaluate the neurodegeneration in HFD mice and AFAD treated.

9. In the discussion section should be stressed the reason why the Authors study the already known effect of this formulation.REF

10. The references Amato A et al. Nutrients 2017 and Nuzzo et al. Curr Alzheimer Res. 2015 should be added in the description of the link between oxidative stress and diseases.

Minor revision:

1.       3. Results line 242 the Author describe a HFD group but a suppose that this is an error.

Author Response

Comments and Suggestions for Authors

In the manuscript, the Authors investigated the neurodegenerative prevention role of ethyl acetate extract from Aruncus dioicus var. kamtschaticus (EFAD) by in vitro and in vivo tests. For in vitro experiments the Authors used EFAD extract in 3T3-L1 cells to prevent lipid accumulation and insulin resistance. The in vivo experiments were performed to demonstrate the EFAD anti-neurodegenerative role in HFD mice. Despite the interesting topic, the manuscript needs specific improvements.

1. During the in vitro tests the Authors used a concentration range of EFAD to demonstrate the reduction of lipid accumulation in 3T3-L1 cells. Could the Authors explain why they didn't use MTS/MTT assay to study cell viability as a function of time and concentration dependence after EFAD administration?

à Thank you for your valuable point.

The cell viability as concentration dependence after EFAD administration was analyzed through MTT assay. Therefore, method, result and figure about this experiment were added to the manuscript.

Line 107-112

2.3.1. Cell viability (MTT assay)

Cell viability was evaluated by MTT assay. 3T3-L1 preadipocytes were plated into 96-well microtiter plates at a density of 1 × 104 cells/well. After 24 h, cells were treated with EFAD for 24 h. MTT solution (5 mg/mL in phosphate buffered saline; PBS) was added to each well and then cells were incubated for 3 h. The medium was removed, and DMSO added to dissolve the MTT formazan. The absorbance was measured at 570 nm using a microplate reader

Line 250-252

The cytotoxicity was confirmed in 200 μg/mL of EFAD, accordingly the experiment proceeded to a maximum concentration 100 μg/mL. (please see attached)

Figure 1. Effects of ethyl acetate fraction from Aruncus dioicus var. kamtschaticus (EFAD) against cytotoxicity, lipid accumulation and insulin resistance on 3T3-L1 cell. (A) Cell viability by MTT assay and degrees of lipid accumulation by Oil Red O staining method. (B) Expression degrees of p-AMPK, p-IRS(ser307), and p-Akt on TNF-α. The values shown are as mean ± SD (n=3; “n” means the technical repeats). The data were statistically considered at p < 0.05 and were represented as different small letters.

2. The Author described a cell increase of p-IRS at ser 307 residue. It is essential to demonstrate that the total insulin receptors remain unchanged. Therefore, I suggest to repeat this western blot whit anti-IRS.

à Thank you for your recommendation.

This result did not show the total IRS, but various factors related to insulin signaling had been identified. In previous study (Ma et al., 2015), the result of comparing p-IRS with β-actin have been published in research journal, in addition, our recent studies including the result of comparing p-IRS with β-actin also published in various journals (Kim et al., 2016; Kim et al., 2017; Kang et al., 2019). One thing is for sure, your opinion is very valuable. Therefore, we will consider it for future experiments. Thank you.

Ma, X.; Fang, F.; Song, M.; Ma, S. The effect of isoliquiritigenin on learning and memory impairments induced by high-fat diet via inhibiting TNF-α/JNK/IRS signaling. Biochem. Biophys. Res. Commun. 2015, 464, 1090-1095.

Kim, J.M.; Park, S.K.; Guo, T.J.; Kang, J.Y.; Ha, J.S.; Lee, U.; Heo, H.J. Anti-amnesic effect of Dendropanax Morbifera via JNK signaling pathway on cognitive dysfunction in high-fat diet-induced diabetic mice. Behav. Brain Res. 2016, 312, 39-54.

Kim, J.M.; Park, C.H.; Park, S.K.; Seung, T.W.; Kang, J.Y.; Ha, J.S.; Lee, D.S.; Lee, U.; Kim, D.; Heo, H.J. Ginsenoside Re ameliorates brain insulin resistance and cognitive dysfunction in high fat diet-induced C57BL/6 mice. J. Agric. Food Chem. 2017, 65, 2719-2729.

Kang, J.Y.; Park, S.K.; Kim, J.M.; Park, S.B.; Yoo, S.K.; Han, H.J.; Kim, D.O.; Heo, H.J. 4, 5dicaffeyolquinic acid improves highfat dietinduced cognitive dysfunction through the regulation of insulin degrading enzyme. J. Food Biochem. 2019, e12855.

3. It’s not completely clear the reason why the Authors use a cell model that is overcome by the experiments in mouse. In fact, in cells they do not investigate the specific pathways involved in reversion of insulin resistance after treatment whit EFAD extract.

à Thank you for your valuable point.

In this study, cell experiments are prior studies for animal experiments. It is well known that p-AMPK, p-IRS (ser) and p-Akt are important factors for insulin action. Therefore, these results demonstrate the potential for improving insulin signaling in EFAD. After confirming the possibility of EFAD through cell experiments, animal experiments were carried out to investigate on improvement effect of EFAD on cognitive impairment by implementing AD characteristics according to peripheral- and CNS-IR.

4. In figure 1 the Authors used n=3. Please indicate what they mean for n. Are technical or biological repeats?

à Thank you for your valuable point.

In Figure 1, “n = 3” means technical repeats, and that mean indicate in the manuscript.

Line 269-270

Expression degrees of p-AMPK, p-IRS (ser307), and p-Akt. The values shown are as mean ± SD (n=3; “n” means the technical repeats).

5. In figure 2(a) the Authors show the mouse body weight after 15 weeks but no information is given for early times. I suggest to start the lines in plot 2a with the initial average body weight of each group. In addition, there is no information provided about the food intake.

à Thank you for your valuable point.

We had checked the initial body weight of mice as well as the feed intake. Therefore, we added the information about initial body weight before eating high fat diets and food intake in the manuscript by adding supplementary material (Table S1).

Line 136-138

The same volume of water was orally administered to the control and HFD group. B.W., food intake and fasting blood glucose from the tail vein were weekly measured.

Line 274-27

There was no significant difference in the initial B.W. of all groups, and the increase of B.W. in the EFAD groups were slower than in the HFD group. However, the food intake was not significantly different between HFD and EFAD groups (Figure 2A, Table S1).

Table S1. Effect of ethyl acetate fraction from from Aruncus dioicus var. kamtschaticus (EFAD) on body weight and food intake of HFD-induced obese mice.

Control

HFD

EFAD20

EFAD40

Initial   body weight (g)

22.46 ± 1.21a

23.00 ± 1.10a

22.91 ± 1.81a

23.64 ± 2.45a

Final body   weight (g)

31.14 ± 1.07c

49.38 ± 2.83a

45.71 ± 2.29b

45.43 ± 2.51b

Food   intake (g/week)

26.18 ± 1.74a

18.35 ± 0.84b

18.18 ± 1.64b

17.73 ± 1.93b

Energy   intake (kcal/week)

100.77 ± 6.71a

96.15 ± 4.40a

95.24 ± 8.60a

92.88 ± 10.11a

The data were statistically considered at p < 0.05 when comparing with the control group and statistical differences were represented as different small letters.

6. In table 1 are shown some markers concerning the liver damage. To correlate this condition to neurodegeneration, I suggest to add the reference Nuzzo D, et al., Nutrients 2018

à Thank you for your valuable suggestion.

We confirm the reference and added the related content from the reference.

Line 500-505

In addition, according to previous research [28], hepatic insulin resistance, oxidative stress and injury together promote the increased generation of “toxic lipids” such as ceramides, which can cross the BBB, they reach the brain and thereby exert neurodegenerative effects via a liver-brain axis. The GOT and GTP as the liver injury factors in the EFAD group were also decreased compared with HFD group. Eventually, these results suggest that EFAD may be effective in improving cognitive impairment due to HFD.

[28] Nuzzo, D.; Amato, A.; Picone, P.; Terzo, S.; Galizzi, G.; Bonina, F.; Mulè, F.; Di Carlo, M. A Natural dietary supplement with a combination of nutrients prevents neurodegeneration induced by a high fat diet in mice. Nutrients 2018, 10, 1130.

7. In figure 6(a) JC-1 was used to measure the mitochondrial stress, but this probe is used in live mitochondria. Are the Authors work whit the live mitochondria? In Addition, in figure 6(b) was used Dichloro-dihydro-fluorescein diacetate (DCFH-DA) assay to estimate the mitochondria oxidative stress but this is not particular for this organelle. I suggest the more specific use of MitoSox.

à Thank you for your valuable point.

We think that the mitochondria is estimated to be a living mitochondria because it is measured immediately after mitochondrial extraction on the day of animal dissection. In the referred reference in our research [23], mitochondrial stress is measured by using the JC-1, and this reference has been cited in numerous articles. Also, the mitochondrial ROS was measured via DCF-DA in this reference [23]. This method is to measure the amount of ROS produced by mitochondria. We sure your suggestion will make our experiments even better. Therefore, we will consider your proposal in future experiments.

[23]. Dragicevic, N.; Mamcarz, M.; Zhu, Y.; Buzzeo, R.; Tan, J.; Arendash, G.W.; Bradshaw, P.C. Mitochondrial amyloid-β levels are associated with the extent of mitochondrial dysfunction in different brain regions and the degree of cognitive impairment in alzheimer's transgenic mice. J. Alzheimer's Dis. 2010, 20, S535-S550.

8. At the end, the neuroprotective effect of EFAD extract requires a stronger evidence. I suggest to make a TUNEL assay to evaluate the neurodegeneration in HFD mice and AFAD treated.

à Thank you for your valuable suggestion.

We conducted a variety of experiments to demonstrate that EFAD has neuroprotective effect. We think that our experimental results can indirectly show that EFAD has neuroprotective effect. Like your suggestion, we also agree that it is important to directly demonstrate neuroprotective effect of EFAD. If we follow your suggestions, we are confident that our research will be even more valuable. However, at present we do not have enough materials to conduct the experiment yet. Therefore, we will reflect your proposal in future experiments through sufficient data research. Thank you.

9. In the discussion section should be stressed the reason why the Authors study the already known effect of this formulation.REF

à Thank you for your valuable point.

In the discussion section, we present the difference between this study and the previous study.

Line 424-428

Through various studies, various physiological activities of Aruncus dioicus var. kamtschaticus in have been proved [11-14]. In animal experiments, it has been confirmed that the Aruncus dioicus var. kamtschaticus is effective for improving diabetes and cognitive function [11, 14]. In this study, we have focused on improvement effect of EFAD on cognitive impairment by implementing AD characteristics according to peripheral- and CNS-IR.

10. The references Amato A et al. Nutrients 2017 and Nuzzo et al. Curr Alzheimer Res. 2015 should be added in the description of the link between oxidative stress and diseases.

à Thank you for your valuable suggestion.

We confirm the referenced and added the related content from the reference.

Line 524-529

Oxidative stress is known to be involved in causing diseases such as non-alcoholic fatty liver disease (NAFLD) and AD. The precise mechanism of the onset and progression of NAFLD remains unclear, but it is suggested that oxidative stress may play a fundamental role [31]. In addition, according to report of Nuzzo et al., HFD induced oxidative stress is associated obesity and AD markers besides insulin resistance [32].

[31] Amato, A.; Caldara, G.; Nuzzo, D.; Baldassano, S.; Picone, P.; Rizzo, M.; Mulè, F.; Di Carlo, M. NAFLD and atherosclerosis are prevented by a natural dietary supplement containing curcumin, silymarin, guggul, chlorogenic acid and inulin in mice fed a high-fat diet. Nutrients 2017, 9, 492.

[32] Nuzzo, D.; Picone, P.; Baldassano, S.; Caruana, L.; Messina, E.; Marino Gammazza, A.; Cappello, F.; Mulè, F.; Di Carlo, M.Insulin Resistance as common molecular denominator linking oesity to Alzheimer’s disease. Current Alzheimer Research 2015, 12, 723-735.

Minor revision:

1.       3. Results line 242 the Author describe a HFD group but a suppose that this is an error.

à Thank you for your point. We change the sentence. Thank you

Line 258-260

The degree of p-AMPK was significantly decreased in TNF-α group (0.77±0.02), and the corresponding value in the control group (1.03±1.00) was confirmed in the EFAD groups (EFAD50, 1.07±0.12; EFAD100, 1.08±0.10, relatively).

Reviewer 2 Report

This is an interesting in vitro study that evaluates the  effects of ethyl acetate fraction from Aruncus dioicus var. kamtschaticus

The findings are  potentially interesting and could represent the basis for further investigations but concerns exist over the methodologic accuracy employed by the Authors and the reported  results .

In addition, please consider the following:

Introduction

The authors should better explain the current knowledge  on  anti-obesity, anto-diabetic, anti-oxidant  activity of Aruncus.

Mat & Met

Generally, It seemed that the authors were not familiar with methods applied in this study. The methodology does not allow  to data interpretation and  a conclusive results

First of all, the in vivo and in  vitro experimental design must be better described.

The culturing conditions and treatments of the  cells described in the different experiments are not clear.

The description of the methods it is very difficult to understand:

Pag .3

line 105  “During the differentiation process, the medium was treated with each concentration of EFAD”,  has no meaning... the cells are treated not the medium

line 112  in “After completion of the differentiation process, the samples were treated for 6 h at different “ samples must be replaced with cells

In the section “western blotting analysis” the authors  must add the protocol, and also the company of all the antibodies used and their working concentrations

Results

In fig 1 there is  a group TNFa  negative and  the group EFAD negative, both goups   have not been clearly defined previously. In addition, the authors say: “The degree of p-AMPK was significantly decreased in HFD group (0.77±0.02), and the corresponding value in the control group (1.03±1.00) was confirmed in the EFAD groups (EFAD50,1.07±0.12; EFAD100, 1.08±0.10, relatively),  but  HFD group is not reported in the  x –axis; and  “the degree of p-IRS(ser307) was significantly increased in the TNF-α group (relative density: 1.00±0.04) and decreased in the EFAD groups (EFAD50, 0.90±0.16; EFAD100, say: 0.77±0.10, relatively), and in particular the EFAD50 group showed a similar degree as the control group (0.80±0.02)” but  Control group is not reported in the  x -axis,

Discussion

The conclusions presented by the authors are not supported by the poor description of the methods and the poor presentation of the results,

Conclusion

From line 544 to558 the authors write the same sentences written  in discussion from line 528 to 542

References

The reference list doesn’t cover the relevant literature adequately and in an unbiased manner. Many errors are present: line 50 Some studies…..only n.6 was reported

Line 192 10 min21, it is the consequence of a cut and a copy?

Line444 Rains.. et al??

In general

the English language  isn’t of sufficient quality, and need to be reviewed throughout the manuscript to eliminate numerous grammar and typographical errors

Author Response

Comments and Suggestions for Authors

This is an interesting in vitro study that evaluates the effects of ethyl acetate fraction from Aruncus dioicus var. kamtschaticus

The findings are potentially interesting and could represent the basis for further investigations but concerns exist over the methodologic accuracy employed by the Authors and the reported results.

In addition, please consider the following:

Introduction

The authors should better explain the current knowledge on anti-obesity, anto-diabetic, anti-oxidant activity of Aruncus.

à Thank you for your valuable suggestion.

These references are the most recent studies in related studies, and we explained the results of these studies in more detail. Thank you.

Line 65-74

Aruncus dioicus var. kamtschaticus is wild herb that is distributed in Northeast Asia such as Korea, Japan and China, recently, a variety of research results have demonstrated the physiological activity of the Aruncus dioicus var. kamtschaticus. In in vitro study, it is reported that extract of Aruncus dioicus var. kamtschaticus protects DNA damage and inhibits the NO activity as well as levels of pro-inflammatory cytokines mRNA expressions in LPS stimulated RAW 264.7 cells [12]. In animal model experiment, it is confirmed that Aruncus dioicus var. kamtschaticus had hypoglycemic and hypolipidemic effects in an animal model of type 2 diabetes [11]. In addition, it was confirmed in a previous study that the ethyl acetate fraction from Aruncus dioicus var. kamtschaticus (EFAD) has an excellent effect on mitochondrial dysfunction based on the superb antioxidant activity of dicaffeoyl glucose isomers [13, 14].

Mat & Met

Generally, It seemed that the authors were not familiar with methods applied in this study. The methodology does not allow to data interpretation and a conclusive results

First of all, the in vivo and in vitro experimental design must be better described.

The culturing conditions and treatments of the cells described in the different experiments are not clear.

The description of the methods it is very difficult to understand:

à Thank you for your valuable point.

We confirmed the contents and corrected the sentence.

Pag .3

line 105  “During the differentiation process, the medium was treated with each concentration of EFAD”,  has no meaning... the cells are treated not the medium

Line 115-116

Each concentration of EFAD were added in the medium during the differentiation process, and the medium was removed after the differentiation process.

line 112  in “After completion of the differentiation process, the samples were treated for 6 h at different “ samples must be replaced with cells

Line 122-123

After completion of the differentiation process, the cells were treated for 6 h at different concentrations (50, 100 μg/mL) of EFAD in the insulin medium.

In the section “western blotting analysis” the authors must add the protocol, and also the company of all the antibodies used and their working concentrations

à Thank you for your valuable suggestion.

The protocol of western blotting is added, and working concentrations were also added in the manuscript. The company of antibodies are already in the manuscript.

Line 86-90

Primary antibodies [β-actin (sc-69879), phosphorylated-insulin receptor substrate (p-IRS)(ser307), phosphorylated-tau (p-tau)(ser404) (sc-12952), Acetylcholinesterase (AChE) (sc-373901), phosphorylated-protein kinase B (p-Akt) (sc-514032) , amyloid-β (Aβ) (sc-28365) and insulin degradation enzyme (IDE) (sc-393887)] were purchased from Santa Cruz Biotechnology (Santa Cruz, CA, USA).

Line 233-242

2.11. Western blot analysis

Liver, white adipose and brain tissue were homogenized with ProtinExAnimal cell/tissue (GeneAll Biotechnology, Seoul, Korea) with 1% protease inhibitor cocktails (Thermo Fisher Scientific, Rockford, IL, USA). The lysates were separated by sodium dodecyl sulfate polyacrylamide gel electrophoresis and transferred to a polyvinylidene difluoride membrane (Millipore, Billerica, MA, USA). The membranes were reacted with primary antibodies diluted with 1:1000 for overnight. And then, the secondary antibody solutions were allowed to react with the membrane for 1 h. Finally, the membranes were examined by exposing the membranes to an ECL reagent (Bionote, Hwaseong, Korea). The luminescence was detected with a ChemiDoc system (iBrightTM CL1000 instrument, Invitrogen, Carlsbad, CA, USA) [17].

Results

In fig 1 there is a group TNFa negative and the group EFAD negative, both goups  have not been clearly defined previously. In addition, the authors say: “The degree of p-AMPK was significantly decreased in HFD group (0.77±0.02), and the corresponding value in the control group (1.03±1.00) was confirmed in the EFAD groups (EFAD50,1.07±0.12; EFAD100, 1.08±0.10, relatively),  but  HFD group is not reported in the  x –axis; and  “the degree of p-IRS(ser307) was significantly increased in the TNF-α group (relative density: 1.00±0.04) and decreased in the EFAD groups (EFAD50, 0.90±0.16; EFAD100, say: 0.77±0.10, relatively), and in particular the EFAD50 group showed a similar degree as the control group (0.80±0.02)” but Control group is not reported in the  x -axis,

à Thank you for your valuable point.

Sorry. there is an incorrect sentence. The HFD group is a TNF-α group.

So, the wrong sentence was corrected, and the figure was also revised.

Line 258-260

The degree of p-AMPK was significantly decreased in TNF-α group (0.77±0.02), and the corresponding value in the control group (1.03±1.00) was confirmed in the EFAD groups (EFAD50, 1.07±0.12; EFAD100, 1.08±0.10, relatively).

please see the attached.

Figure 1. Effects of ethyl acetate fraction from Aruncus dioicus var. kamtschaticus (EFAD) against cytotoxicity, lipid accumulation and insulin resistance on 3T3-L1 cell. (A) Cell viability by MTT assay and degrees of lipid accumulation by Oil Red O staining method. (B) Expression degrees of p-AMPK, p-IRS(ser307), and p-Akt on TNF-α. The values shown are as mean ± SD (n=3; “n” means the technical repeats). The data were statistically considered at p < 0.05 and were represented as different small letters.

Discussion

The conclusions presented by the authors are not supported by the poor description of the methods and the poor presentation of the results,

à Thank you for your valuable point.

We have added some content and references.

Line 424-428

Through various studies, various physiological activities of Aruncus dioicus var. kamtschaticus in have been proved [11-14]. In animal experiments, it has been confirmed that the Aruncus dioicus var. kamtschaticus is effective for improving diabetes and cognitive function [11, 14]. In this study, we have focused on improvement effect of EFAD on cognitive impairment by implementing AD characteristics according to peripheral- and CNS-IR.

Line 500-505

In addition, according to previous research [28], hepatic insulin resistance, oxidative stress and injury together promote the increased generation of “toxic lipids” such as ceramides, which can cross the BBB, they reach the brain and thereby exert neurodegenerative effects via a liver-brain axis. The GOT and GTP as the liver injury factors in the EFAD group were also decreased compared with HFD group. Eventually, these results suggest that EFAD may be effective in improving cognitive impairment due to HFD.

[28] Nuzzo, D., Amato, A., Picone, P., Terzo, S., Galizzi, G., Bonina, F., ... & Di Carlo, M. (2018). A natural dietary supplement with a combination of nutrients prevents neurodegeneration induced by a high fat diet in mice. Nutrients, 10(9), 1130.

Line 524-529

Oxidative stress is known to be involved in causing diseases such as non-alcoholic fatty liver disease (NAFLD) and AD. The precise mechanism of the onset and progression of NAFLD remains unclear, but it is suggested that oxidative stress may play a fundamental role [31]. In addtion, according to report of Nuzzo et al., HFD induced oxidative stress is associated obesity and AD markers besides insulin resistance [32].

[31] Amato, A.; Caldara, G.; Nuzzo, D.; Baldassano, S.; Picone, P.; Rizzo, M.; Mulè, F.; Di Carlo, M. NAFLD and atherosclerosis are prevented by a natural dietary supplement containing curcumin, silymarin, guggul, chlorogenic acid and inulin in mice fed a high-fat diet. Nutrients 2017, 9, 492.

[32] Nuzzo, D.; Picone, P.; Baldassano, S.; Caruana, L.; Messina, E.; Marino Gammazza, A.; Cappello, F.; Mulè, F.; Di Carlo, M.Insulin Resistance as common molecular denominator linking oesity to Alzheimer’s disease. Current Alzheimer Research 2015, 12, 723-735.

Conclusion

From line 544 to558 the authors write the same sentences written in discussion from line 528 to 542

à Thank you for your valuable point.

Sorry. we deleted all repeated sentences in the discussion section line 528 to 542. Thank you.

References

à Thank you for your valuable point.

The wrong sentence was confirmed and corrected.

The reference list doesn’t cover the relevant literature adequately and in an unbiased manner. Many errors are present: line 50 Some studies…..only n.6 was reported’

Line 48-50

Recently, it has reported that chronic hyperglycemia and hyperinsulinemia due to peripheral-IR cause central nervous system (CNS)-IR, which in turn results in cognitive impairment in the brain, leading to Alzheimer's disease (AD) [6].

Line 192 10 min21, it is the consequence of a cut and a copy?

Line 202-203

Absorbance was measured at 405 nm at intervals of 1 min for 10 min [21].

Line444 Rains.. et al??

Line 466-468

According to a study by Rains et al., the phenolics present in foods are known to have anti-obesity effects by promoting energy consumption and lipid oxidation, inhibiting appetite, delaying nutrient absorption, and alternating in fat metabolism [10].

In general

the English language isn’t of sufficient quality, and need to be reviewed throughout the manuscript to eliminate numerous grammar and typographical errors.

à We have already been reviewed for English language. We attached the document to prove that this manuscript has been reviewed by a native speaker.

Round 2

Reviewer 1 Report

The manuscript is acceptable for publication in its present form